# Critical Test Result Recall Supporting System (CTR RSS) Improves Follow-Up among Patients in the Community

**DOI:** 10.3390/diagnostics12051252

**Published:** 2022-05-18

**Authors:** Hsu-Tung Chang, Su-Feng Kuo, Shu-Hui Chen, Jen-Shiou Lin, Shu-Hui Lin, Chin-Fu Chang, Chih-Wen Twu, Mei-Chu Chen, Yuan-Ting Yang, Chew-Teng Kor, Ching-Hsiung Lin

**Affiliations:** 1Department of Hospital Quality Management, Changhua Christian Hospital, Changhua 500, Taiwan; 60452@cch.org.tw (H.-T.C.); 182737@cch.org.tw (C.-W.T.); 2Department of Laboratory Medicine, Changhua Christian Hospital, Changhua 500, Taiwan; 60368@cch.org.tw (S.-F.K.); 97688@cch.org.tw (J.-S.L.); 3Endoscopy Center, Changhua Christian Hospital, Changhua 500, Taiwan; 6484@cch.org.tw; 4Department of Surgical Pathology, Changhua Christian Hospital, Changhua 500, Taiwan; 74630@cch.org.tw; 5Department of Emergency Medicine/Critical Care, Changhua Christian Hospital, Changhua 500, Taiwan; 68358@cch.org.tw; 6Department of Otorhinolaryngology-Head and Neck Surgery, Changhua Christian Hospital, Changhua 500, Taiwan; 7Nursing Department, Changhua Christian Hospital, Changhua 500, Taiwan; 16749@cch.org.tw; 8Pharmacy Department, Changhua Christian Hospital, Changhua 500, Taiwan; 156393@cch.org.tw; 9Big Data Center, Changhua Christian Hospital, Changhua 500, Taiwan; 179297@cch.org.tw; 10Graduate Institute of Statistics and Information Science, National Changhua University of Education, Changhua 500, Taiwan; 11Division of Chest Medicine, Department of Internal Medicine, Changhua Christian Hospital, Changhua 500, Taiwan; 12Institute of Genomics and Bioinformatics, National Chung Hsing University, Taichung 402, Taiwan; 13Department of Recreation and Holistic Wellness, MingDao University, Changhua 523, Taiwan

**Keywords:** critical test results (CTR), patient in the community, CTR recall support system, return follow-up

## Abstract

Follow-up care of patients in the community is an important topic for improving patient outcomes, especially when medical personnel receives a notification of the critical test result (CTR) when the CTR becomes available after patients have been out of hospital; how to recall the patient back to the hospital and follow-up treatment is essential for preventing the healthcare risk of neglecting or delayed intervention with respect to the patient’s CTR. We are concerned that the follow-up of CTR and timely recall of our patients in the community improves and facilitates patient safety. We built the CTR Recall Supporting System (RSS) to follow up and recall our patients in the community. Measures were introduced to evaluate the effectiveness of CTR RSS; the rate of return of patients within 7 days increased from 58.5% to 88.8%, an increase of 30.3%, the patients in the community’s return follow-up interval days decreased from 10.9 days to 6.2 days, reduced by 4.7 days (*p* < 0.001), and the mortality rate of the patients in the community within 48 h decreased from 8.0% to 1.9%, a decrease of 6.1%, *p* < 0.001. The implementation of the CTR RSS significantly increases the discharged patient in he community’s CTR return follow-up within 7 days rate, decreases CTR return follow-up interval days, and reduces the CTR mortality rate within 48 h. This effectively improves the effects of CTR on return follow-up visits and provides a prototype system for hospitals that intend to improve this issue.

## 1. Introduction

The critical test result (CTR) refers to variability that deviates from the normal range, indicating a high-risk or life-threatening condition of an urgent nature (e.g., severe hypoglycemia and hyperkalemia revealed by tests, aortic dissection, and pneumothorax revealed by radiology), for which immediate medical action must be taken to protect life or prevent the occurrence of complications [1].

In 2005, Massachusetts hospitals joined the Massachusetts Coalition for the Prevention of Medical Errors to have a patient safety initiative to improve effective CTR communications (in a timely and accurate manner). According to the recommendations of the Coalition-convened consensus group, CTR items include laboratory, cardiology, radiology, and other diagnostic tests, as well as developed Safe Practice Recommendations, such as the scope of CTR, who receives CTR, how to inform and remind physicians, how to ensure that physicians receive correct reports in time, the CTR intervention records, and how to inform patients of their lab results and follow-up treatments [2].

However, systematic reviews of test result follow-ups have shown that pathology and imaging test results are not followed up for 20–62% of inpatients and for up to 75% of patients treated in an emergency department (ED). Sydney Hospital showed that tests ordered on the day of discharge from the patient accounted for 47% of missed test results. Poor follow-up of the test results can have major consequences on the quality of care [3].

The period after discharge is a vulnerable time for patients; 41% of patients left the hospital before all laboratory and radiological test results were finalized. Of these results, 9.4% were potentially actionable and could have altered management. Physicians were aware of only 38% of post-discharge test results, so there is a need to track and alert providers of the finalized post-discharge test results. Electronic applications have been developed to support test-result management processes, including tracking pending test results at hospital discharge, delivering result alerts to clinicians, acting as safety nets in result notification, or using tracking systems to document acknowledgment and clinical actions [4].

The National Clinical Program for Pathology has developed a guide for the communication of critical laboratory results to patients in the community to provide clear information for both laboratories and healthcare professionals in relation to minimum recommendations for the communication of critical laboratory results. It is recommended that the demographic data of the patients collected should include the mobile phone number of the patient and/or their caregiver/next relatives to facilitate contact with the patient in the event that a critical result is obtained in the analysis of a test and the healthcare professional, or their nominee, cannot be contacted [5].

Many hospitalized patients have pending microbiological test results at the time of discharge. Failure to follow up on these results in a timely manner can lead to delays in diagnosis and adequate treatment of important infections. Many microbiological results return after discharge, and some require a change in treatment [6].

Recall is an important mediating variable for improved treatment adherence and health outcomes. Factors that influence recall are the patient, information, and communication [7]. When patients are in the community, how to timely inform the patient and recall them to treatment is a challenge for medical staff to follow. For our hospital, CTR becomes available after patients have left the hospital. The CTR return follow-up rate within 7 days was only 58.5%, indicating that there is a risk of lost or delayed intervention of CTR for patients outside of the hospital. In this study, we want to build CTR RSS to follow-up and timely recall our patients in the community and evaluate the effectiveness of it to improve and facilitating patient safety.

## 2. Materials and Methods

### 2.1. Setting of the Study

Hospital scale: Central Taiwan Medical Center, with 1655 beds, tertiary hospital affiliated with six regional branch hospitals. Sixty CTRs have been listed based on the risks of clinical operation requirements (e.g., severe hyperglycemia or hypoglycemia (glucose 600 mg/dL or 40 mg/dL), hyperkalemia or hypokalemia (K 6.2 mmol/L or ≤2.5mmol/L); aortic dissection, and pneumothorax as revealed by radiology). In 2019, there were 1,514,316 outpatient visits, 48,638 discharges, 80,250 emergency visits, and CTR notifications for 9336 patients (52.6% inpatients, 5.5% outpatients, and 41.9% emergency visits).

### 2.2. The CTR System

The hospital’s interdisciplinary Quality Committee for Clinical Testing and Examinations is responsible for the coordination of various departments (namely, the laboratory department, pathology, radiology, units performing point-of-care testing, nursing department, information, physicians, and other units relevant to physiological examinations) for CTR notification, reviewing CTR items, clarifying notification of alert threshold, and monitoring of the effectiveness of the notification process. The hospital’s CTR notification process flowchart is shown in Figure 1.

Regarding the CTR notification process of patients, the hospital changed its communication devices in January 2019, from the former SMS one-way notification of the mobile phone to the internal software on the communication platform (M + Messenger) smartphone, certified by ISO 27001 and ISO 27011 international standards for information security. The platform saves the cost of sending SMSs, provides two-way feedback, displays examination images, and queries reports. The physician in charge must administer the intervention within 30 min after receiving the notification and have an intervention record linked to the integrated electronic medical record of the patient, including medication, treatment of the procedure, observation, transfer, or arrangement of follow-up visits. Lab staff monitor CTR physician feedback and intervention through the CTR Audit System. When physicians receive the CTR notification, and if the patient is in the hospital, they will provide intervention immediately; but, if the patient has been in the community, they must recall the patient for intervention to reduce the risk of conditions. The CTR Audit System shows that the acknowledgment within 30 min and the intervention rate of the CTR notifications reached 100% and 97.6%, respectively. However, when the CTR became available after patients were out of the hospital, the CTR follow-up process tracked 443 CTR patients in the community from July to December 2019; the patient in the community CTR return follow-up rate within 7 days was only 58.5%.

### 2.3. Study Design

A retrospective review design was used, and the data source was derived from the CTR Audit System. To collect data, four members were involved: a laboratory medicine physician, imaging paramedics, a quality manager, and a nurse supervisor. These members are members of the quality committee for clinical tests and examinations and are well-trained and experienced in CTR data collection. Data collection included patient gender, responsible physician, physician departments, care area (outpatient, emergency, inpatient), in community, CTR items, CTR notification time, time appointed follow-up visit, follow-up visit time, and patient’s death. The study was necessary to retrieve patient data to track the CTR intervention of the patients; this study had obtained the clinical trial certification (IRB number 201023) from the Institute Research Board (IRB) of Changhua Christian Hospital. As the study involved only medical records, which made the study protocol a lower risk, the risks for the study subjects are not higher than those of the non-subjects. Therefore, the IRB granted an exemption for informed consent.

### 2.4. Study Process

The target group is the patients in the community when CTR becomes available after patients have been out of the hospital from July 2019 to April 2021. The hospital completed the conversion of communication devices in July 2019, but the CTR RSS had not been activated. Taking into account the completeness and validation of the data, we used July to December 2019 as the 6-month observation period for the baseline (usual group). CTR RSS was launched in January 2020, with a 16-month implementation period (reinforce group) from January 2020 to April 2021; patient death discharges were excluded from the groups. We compared the two groups on return and unreturning follow-up measures (as shown in Figure 2).

### 2.5. Intervention Protocol

As mentioned above, recall is an important mediating variable for improved treatment adherence and health outcomes; factors influencing recall are patient information and communication. Therefore, we want to manage CTR RSS for follow-up and recall of our patients in the community in a timely manner and evaluate the effectiveness of it to improve and facilitate patient safety (as shown in Figure 1). The CTR RSS for patients in the community performs as follows:

Link comprehensive follow-up information for patients in the community: The responsible physician receives CTR but is not noted if the patient is in the community, and cannot distinguish whether the patient is out of hospital; even if the physician knew that the patient had been in the community, the CTR notification did not have the patient’s contact information, therefore, the physician contacts the patient inconveniently. Therefore, the CTR RSS for patients in the community provides information on clinical diagnosis, laboratory data, radiology images, treatment, and patient medications to physicians. It also notes whether the patient has been in the community and includes the contact information of the patient or their carer/next relatives, making it easier for physicians to contact patients and shortening the time required to inquire for patient information. After recalling, the system notes the return visit time that has been confirmed with the patient. If the patient did not return to the hospital on time, the system will remind the health care team to follow up.

Engaging a responsible medical care team for recalling patients: Regarding the recall of patients in the community to return to the hospital for a CTR intervention, the responsibilities of the recall of patients for return follow-up are not clearly defined; therefore, there will be patients who have not been recalled or patients who have been notified but did not return to the hospital. As a result, the tracking of patients by responsible physicians has been standardized, and the supporting personnel of the responsible medical care team (e.g., the chief resident, the on-duty physician, the nursing specialist, or the case manager who handles patient diseases, such as chronic kidney disease, acute myocardial infarction, heart failure, or asthma) would be initiated by responsible physicians. They are responsible for recalling patients, explaining to them the reason for the recall, the progress of their condition, and possible interventions. As the responsible medical care team is familiar with the patient’s condition, they are able to clearly instruct the patient to return to the hospital for an intervention, improving the willingness of the patient to return to the hospital for follow-up and intervention.

Creating an online CTR consultation platform for patients: Since the patient cannot learn the CTR after leaving the hospital, to ensure the correctness and timeliness of the CTR recall communication for patients in the community, we provide a customized two-way consultation platform for the CTR network of patients in the community. On their smartphones, patients can inquire about their CTR, its trends and warnings and ask questions about their tests. In this way, patients engage in communication with the responsible medical care team and the patient is informed, which also increases the willingness of the patient to return to the hospital for intervention.

### 2.6. Outcome Variables

The effectiveness of CTR recalling patients in the community to the hospital includes the following aspects: patients in the community CTR return follow-up rate within 7 days rate, patients in the community CTR return follow-up interval days, and patients in the community CTR mortality rate within 48 h.

### 2.7. Statistical Analysis

Descriptive data were represented as numbers (percentage) and the mean (standard deviation) for categorical and continuous data, respectively. Categorical data were compared between the usual and reinforced groups using the chi-square test. Student’s *t*-test was used for continuous data comparison. Segmented time-series regression analyses were performed to evaluate the effectiveness of the interventions. The relative risk (RR) and the 95% confidence interval (CI) were calculated and provided evidence of an increase or reduction in clinical outcome after intervention. All statistical analyses were performed using R software and the “TSMODEL” R package for the Poisson segment regression model. Two-sided *p*-values of <0.05 were considered statistically significant.

## 3. Results

The target group has 1482 patients in the community from July 2019 to April 2021 when CTR becomes available after patients have left the hospital. With respect to the 443 patients in the community (usual group), the patient in the community CTR return follow-up within 7 days rate was only 58.5% and non-return was 41.5%. In terms of care area, emergency was 37.2%, inpatient was 36.4%, and outpatient was 26.4%. In terms of CTR elements, blood culture ranked the highest (69.2%), followed by potassium (7.6%), magnetic resonance imaging (MRI) (3.2%), computed tomography (CT) (2.7%), sodium (2.7%), X-ray (2.2%), and others (glucose, pH, and cardiac ultrasound), which accounted for 12.4% in total, as shown in Table 1.

### 3.1. Comparison of Outcome Measures between the Usual Group and Reinforce Group

For the usual group, 443 patients in the community, the patients in the community CTR return follow-up rate within 7 days was only 58.5%, while the reinforced group’s, 1039 discharged patients, the rate was 88.8%, an increase of 30.3% (95% confidence interval, CI: 29.2–32.4%, *p* < 0.001), as shown in Figure 3 and Table 2. The days of community return follow-up of the CTR returned patients ranged from 10.9 days (usual group) to 6.2 days (reinforce group), a reduction of 4.7 days (95% CI, 3.9–5.7, *p* < 0.001), as shown in Figure 4 and Table 2. When CTR becomes available after patients have been out of the hospital, the mortality rate of community CTR patients in 48 h ranged from 8.0% (usual group) to 1.9% (reinforce group), a decrease of 6.1% (95% CI, 6.0–8.5%, *p* < 0.001), as shown in Figure 5 and Table 2.

### 3.2. Comparison of CTR Un-Return Items Contemporary Comparison between Usual Group and Reinforced Group

When the percentages of patients in the community who did not return for the CTR items were compared, there was a drop of 30.6% from 41.9% of the usual group (July–December 2019) to 11.3% of the contemporary reinforced group (July–December 2020). Of the CTR items (blood culture, potassium, MRI, etc.) of discharged patients who did not return for treatment, blood culture ranked the highest; the percentages ranged from 56.4% to 17.8%, a drop of 38.6%. There were also decreases in outpatient (23.4% vs. 3.4%), emergency (47.6% vs. 14.2%), and inpatient care areas (75.3% vs. 28.9%). All of these measures had a significant improvement, *p* < 0.001, respectively, as shown in Table 3.

## 4. Discussion

The implementation of the CTR RSS significantly increases the patient in the community CTR return follow-up rate within 7 days and decreases the patient in the community CTR return follow-up interval days and patients in the community CTR mortality rate within 48 h. In terms of CTR un-returned item contemporary comparison, the numbers of those cases in the outpatient, emergency, and inpatient care areas were also significantly improved. This effectively improves the effects of CTR on return follow-up visits and provides a prototype system for hospitals that intend to improve this issue.

By using CTR RSS, the physician is able to determine whether or not a patient in the community needs a recall based on their critical result, disease condition, and history in order to prevent unnecessary recall. To ensure timely and patient-centered care with the help of the recall team, our recall team actively reaches out to patients according to contact information to prevent them from ignoring the CTR notification or message and assure their need for assistance in re-visit if they need it. The recall team also monitors re-visits to ensure they adhere to follow-up. In addition, doctors must make sure to respond to physician feedback within 30 min of receiving it. In order to prevent them from ignoring, the recall team will contact a physician if they did not meet the requirements.

Information technology has the potential to enhance the performance and safety of test result management processes [8]. Therefore, we provide information on clinical diagnosis, laboratory (examinations), treatment, and medications of patients in the community to physicians, and through the internal software communication platform (M + Messenger) smartphone, we provide convenient two-way feedback notifications, as well as the information of intervention and diagnostics (including imaging), improving the accuracy and timeliness of the CTR process. Physicians receive the notification and determine whether or not to recall the patient regarding not only the CTR but also disease conditions, medical history, examination, treatment, medication, and other information. This mechanism is aimed to prevent unnecessary recall and re-visit. For example, although emergency cases of hyperkalemia are usually acute and life-threatening, for chronic diseases in the Department of Endocrinology or Nephrology (such as CKD), it would be an expected or non-urgent CTR, and the patient does not necessarily need to return to the hospital for follow-up intervention. Therefore, the initiation of a return follow-up CTR in the hospital must be customized according to the conditions of the disease to help the responsible medical care team make rapid and appropriate clinical CTR decisions to treat patients in the community [9].

When laboratory tests or examination reports meet these thresholds, they are considered CTRs, then physicians receive the notification and determine whether or not to recall the patient regarding not only the CTR but also disease conditions, medical history, examination, treatment, medication, and other information. This mechanism is aimed at preventing unnecessary recall and re-visit. For example, despite emergency hyperkalemia cases being acute and life-threatening, for chronic diseases in the Departments of Endocrinology and Nephrology (i.e., CKD), it may be an expected or non-urgent CTR, and the patient does not necessarily need to return to the hospital for follow-up treatment.

We audit the outcome measures in those patients to reflect the effectiveness of CTR return follow-up in the patients in the community. It is necessary to further follow whether the physician has provided the intervention in accordance with clinical guidelines, such as changes in warfarin dosing and PT/INR [10]. Studies may further track the appropriateness of blood culture and potassium intervention.

Regarding the MRI, CT, and X-ray examination reports, although the proportion of not returning to the hospital for follow-up was not as high (approximately 8.1%), an improvement was observed through the CTR RSS, but it was not significant (*p* > 0.001). Failure to return to the hospital for follow-up of imaging reports was found primarily in the outpatient care area because imaging required interpretation by the radiology and radiology physicians and interdisciplinary communications, which would be time-consuming. Outpatient visits were often short, and it was difficult to complete the report before the patients left the hospital. Therefore, if information technology, such as artificial intelligence to interpret images, is introduced to the outpatient care area, it may help to shorten the time to image CTR reports, completing the report before the patient leaves the hospital, helping to reduce the difficulty of tracking patients in the community [11]. This study found that the imaging reports of failure to return to the hospital for follow-up were mainly found in the outpatient care area, where outpatient visits were often short; it was difficult to complete the report before the patient left the hospital because the imaging required human interpretation and interdisciplinary communications, which was time-consuming. With reference to the previous studies mentioned above, the application of artificial intelligence to the outpatient lab (examinations) is of critical importance to the CTR RSS and may be used as a future improvement strategy.

It is also important to improve the laboratory (examinations) process (preanalytic, analytic, and postanalytic). We found that there were epidermal bacteria in certain blood cultures, which was inconsistent with the patient’s symptoms, and the patient’s symptoms had improved after treatment. In reviewing the process, it was caused by sample contamination, and the patient would not have to return to the hospital for treatment. Previous studies found that by reducing non-disease factors, such as incorrect sample collection methods, storage errors, delivery delays, or transcription errors, unexpected CTR may be reduced as well [12,13]. In the future, an important improvement measure is to reduce the incidence of CTR as a way to avoid unnecessary follow-up visits from patients in the community.

The limitation of this study was that it focused mainly on CTR. Abnormal results, such as test or examination results that have no immediate significant effect on patient vital signs but may change clinical treatment, such as hemoglobin level of ≥20 g/dL or ≤6 g/dL, were not included in the scope of the study. Regarding the 48-h CTR mortality rate, the mortality rate of patients in the community who returned to the hospital within 48 h after the CTR notification was issued and died, those who died in other hospitals or their own homes were not included in the scope of this study measure because it was difficult to evaluate the correlation with the CTR. Measures that are not standardized and adjusted for disease type or patient severity, as well as the effect of CTR, change mortality [14]. In addition, we clarified items for critical examination reports, and the CTR Audit System may contribute to preventing and reducing missing data or incomplete data. Actually, critical test results are identified automatically by comparing digital numbers to a threshold; therefore, there is no missing data. In contrast, examination reports relying on free-text contents are seldom automatic, leading to missing information; this is the limitation of our CTR. Recently, automatic summarization of the free text content using artificial intelligence (AI) in care episodes has been reported; this could assist clinicians [15]. We suggest that the introduction of AI technology could be helpful in reducing missing data.

## 5. Conclusions

This study found that the implementation of the CTS RSS significantly improves the outcome measures of CTR. The decrease in the number of patients who did not return for a follow-up in the outpatient, emergency, and inpatient care areas contributed mainly to the provision of comprehensive information on patients in the community to the responsible medical care team and to the follow-up of patient return follow-up. On the patient’s end, the provision of CTR trend information and return notification instructions helped to facilitate communication between the responsible medical care team and the patient. It may also provide a prototype system for hospitals that intend to improve the return follow-up of patients in the community. However, CTR clinical decision-making must still be based on reviewing disease conditions, medical history, examination, treatment, medication, and other information. The CTR initiation of return follow-ups must also be customized to avoid the burden of the recall operation, as well as lead patients to return to the hospital inconveniently and unnecessarily.

## Figures and Tables

**Figure 1 diagnostics-12-01252-f001:**
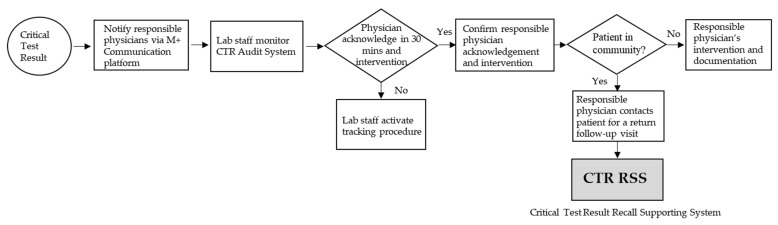
Hospital CTR Processing Flowchart.

**Figure 2 diagnostics-12-01252-f002:**
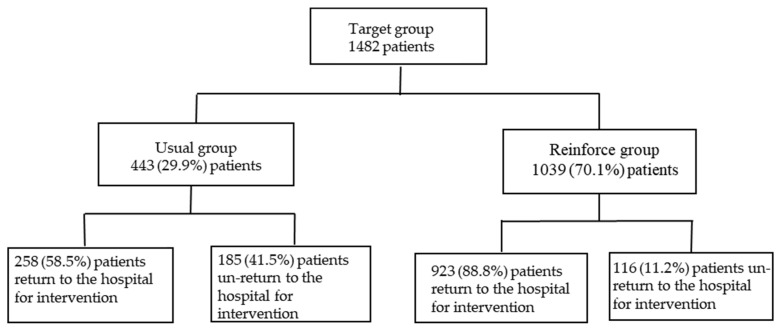
Target, usual, and reinforced group distribution of the numbers of patients in the community (excluded patient deaths). Target group: patients in the community when CTR becomes available after patients have been out of hospital from July 2019 to April 2021; Usual group: CLS RSS was not initiated in July–December of 2019; Reinforce group: CLS RSS was initiated in January 2020–April 2021.

**Figure 3 diagnostics-12-01252-f003:**
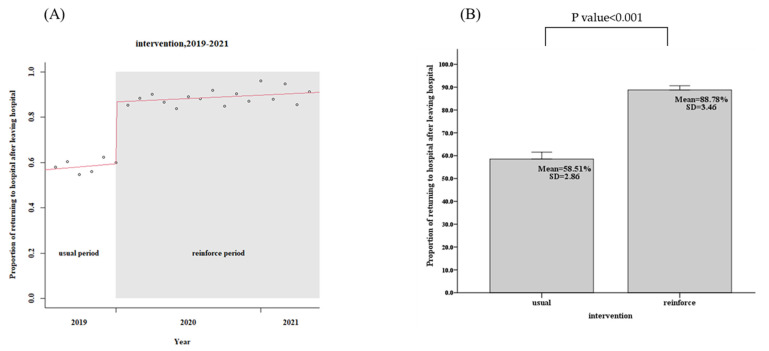
Patients in the community CTR return follow-up rate within 7 days (usual group vs. reinforced group). (**A**) Trends, (**B**) Average values.

**Figure 4 diagnostics-12-01252-f004:**
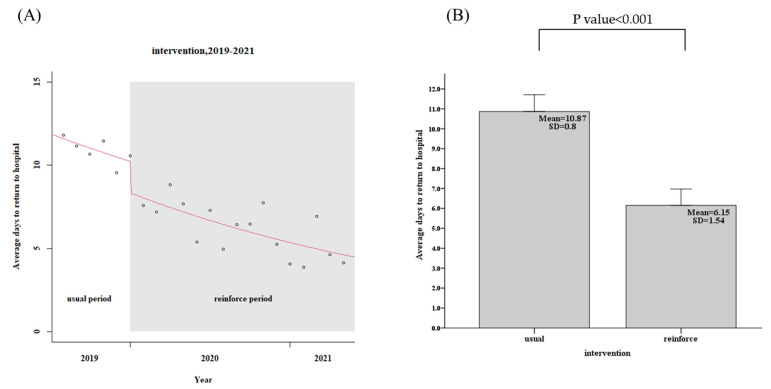
Patients in the community CTR return follow-up interval days (usual group vs. reinforced group). (**A**) Trends, (**B**) Average values.

**Figure 5 diagnostics-12-01252-f005:**
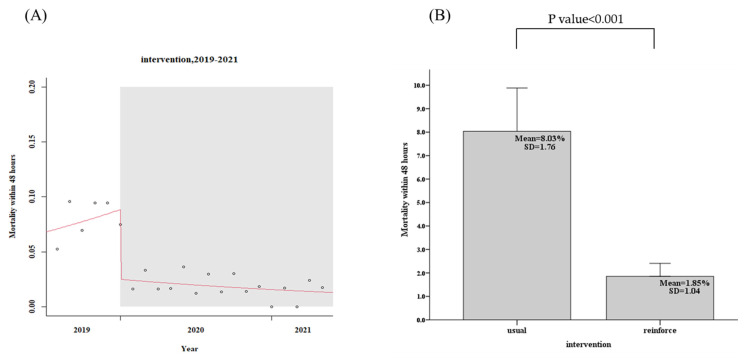
Patients in the community CTR mortality rate within 48 h (usual group vs. reinforced group). (**A**) Trends, (**B**) Average values.

**Table 1 diagnostics-12-01252-t001:** CTR items for patients in the community who did not return for a follow-up between July and December 2019.

CTR Items	Care Areas			Number of Cases	Ratio
	Outpatient	Emergency	Inpatient		
Blood culture	3	64	61	128	69.2%
Potassium	13	0	1	14	7.6%
MRI	6	0	0	6	3.2%
Sodium	4	1	0	5	2.7%
CT	2	3	0	5	2.7%
X-ray	2	1	1	4	2.2%
Other	19	0	4	23	12.4%
Total	49	69	67	185	100.0%
Ratio	26.4%	37.2%	36.4%	100.0%	-

**Table 2 diagnostics-12-01252-t002:** Patients in the community CTR return follow-up rate within 7 days, patients in the community CTR return follow-up interval days, and patients in the community CTR mortality rate within 48 h. (usual group vs. reinforced group).

CTR Outcome Measures	Usual Group (*n* = 443) (%)	Reinforce Group (*n* = 1039) (%)	Differences (%) (95% CI)	*p*-Value	Usual Group			Reinforce Group		
					RR *	CI	*p*-Value	RR	CI	*p*-Value
Follow-up of the CTR return within 7 days	58.2	88.8	30.3 (29.2–32.4)	<0.001	1.01	0.94–1.08	0.831	0.99	0.93–1.07	0.894
CTR return follow-up interval days	10.9 days	6.2 days	4.7 days (3.9–5.7)	<0.001	0.98	0.96–0.997	0.023	0.98	0.97–1.01	0.286
CTR mortality rate within 48 h	8.0	1.9	6.1 (6.0–8.5)	<0.001	1.05	0.86–1.27	0.653	0.92	0.74–1.14	0.454

* RR: Relative risk.

**Table 3 diagnostics-12-01252-t003:** Comparison of CTR items for community patients who did not return for a follow-up between the usual group (July–December 2019) and the contemporary reinforced group (July–December 2020).

CTR Items	Outpatient		Emergency		Inpatient		Usual			Reinforce			Differences	*p*-Value
	Usual	Reinforced	Usual	Reinforced	Usual	Reinforced	Un-return	Recall	Ratio	Un-return	Recall	Ratio		
Blood culture	3	0	64	11	61	21	128	227	56.4%	32	180	17.8%	38.6%	<0.001
Potassium	13	1	0	2	1	0	14	81	17.3%	3	82	3.7%	13.6%	0.010
MRI	6	2	0	1	0	0	6	7	85.7%	3	8	37.5%	48.2%	0.119
Sodium	4	1	1	0	0	0	5	13	38.5%	1	11	9.1%	29.4%	0.166
CT	2	0	3	1	0	0	5	6	83.3%	1	7	14.3%	69.0%	0.029
X-ray	2	1	1	0	1	0	4	6	66.7%	1	5	20.0%	46.7%	0.242
Other	19	2	0	0	4	1	23	102	22.5%	3	97	3.1%	19.4%	<0.001
Un-return	49	7	69	15	67	22	185	442	41.9%	44	390	11.3%	30.6%	<0.001
Recall	209	207	145	106	89	76	-	-	-	-	-	-	-	-
Ratio	23.4%	3.4%	47.6%	14.2%	75.3%	28.9%	-	-	-	-	-	-	-	-
*p* Value	<0.001		<0.001		<0.001		-	-	-	-	-	-	-	-

## Data Availability

Data cannot be provided with the paper but can be made accessible on individual request.

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
