# Peer review of "Critical Test Result Recall Supporting System (CTR RSS) Improves Follow-Up among Patients in the Community"

_diagnostics, 2022, doi:10.3390/diagnostics12051252_

Round 1

Reviewer 1 Report

Firstly, thank you for opportunity to review very interested article. I don't feel qualified to judge about the English language and style due to not native language.

  1. The title reflect the main subject about critical test result recall supporting system, title was clear and easy to understand. 
  2. The abstract summarize and reflect the work described in the manuscript.
  3. The key words reflect the focus of the manuscript.
  4. The manuscript adequately describe the background, present status, and significance of the study. The authors explain recall supporting system in many countries around the world. However, I suggested the authors to more clarify about benefit to set up this system and type of test which need to used this system.
  5. The manuscript describe methods in adequate detail, study subjects were clear, with demonstrate IRB number or text to human ethics consideration. I suggest the authors explain in detail about                         - Who's response for data collection? (number of collector, qualify)       - Data validation in case of missing data or incomplete data.            
  6. The research objectives achieved by the experiments used in this study.
  7. The manuscript interpret the findings adequately and appropriately, highlighting the key points concisely, clearly, and logically.
  8. Tables and figures sufficient, good quality and appropriately illustrative of the paper contents.
  9. The manuscript meet the requirements of biostatistics.
  10. The manuscript cite appropriately the latest, important, and authoritative references in the introduction and discussion sections. However, some of references were incorrect style for this journal.

Author Response

We sincerely appreciate reviewer taking time to provide their valuable comments and suggestion, which helped us to improve the quality of the article. We have addressed the Reviewers' comments point by point. Please see the attachment.

Reviewer 2 Report

To build up a CRI and RSS system is a great Idea, and it surely will improve patients safety  under appropriate conditions. A system like this will not be easily used generally, and the effectiveness will depend on the local health system conditions, e.g., which information is given to the patient when leaving the hospital.

Limitations:

Data protection rules may be handled differently in different countries, and this might limit the general usability of the system, which may be a part of the limitations

The National Clinical Program for Pathology is edited by the Royal College of Physicians of Ireland.  The authors describe that the registration for testing requires a data sharing agreement comply with General Data Protection Regulations (GDPR), or part of a Service Level Agreement (SLA). The General data protection Regulations are under the Regulation of locally applicable law.

The system tested by the study design and the results depend on the local conditions of medical care in Taiwan. The follow- up statistics with major consequences are derived from Australian conditions, the rate of incomplete findings after leaving the hospital with 41% as well. It is unclear, how the patients reacted on the finding of a CRI. It can be ignored, or other medical institutions can be contacted, or when there are big distances to be overcome, the re-visit may be dropped.

The paper does not include the acuity of the situation, which led to the tests, providing critical results. So, the number of critical test results might even reflect a medical decision making based on incomplete reviewing of disease conditions, medical history, examination, treatment, medication, and other information. (See line 320), a topic worth being more extensively discussed, and might serve as a quality indicator.  

This study is based on test parameters without reflecting the clinical conditions. Nevertheless, it shows the  significant improvement of medical care by controlling omly test parameters and recalling patients with critical parameters for a follow-up.  

Author Response

(The authors gave the same response as above.)
